



# Investigating the zero transmission problem in satellite solar occultation measurements in the context of possible stratospheric aerosol injections

Anna Lange [1], Ulrike Niemeier [2], Alexei Rozanov [3], and Christian von Savigny [1]

[1]Institute of Physics, University of Greifswald, Felix-Hausdorff-Str. 6, 17489 Greifswald, Germany
[2]Max Planck Institute for Meteorology, Bundesstr. 53, 20146 Hamburg, Germany
[3]Institute of Environmental Physics, University of Bremen, Otto-Hahn-Allee 1, 27359 Bremen, Germany

**Correspondence:** Anna Lange (anna.lange@uni-greifswald.de)

**Abstract.** Stratospheric aerosol injections have been proposed to mitigate the effects of global warming. The injection of sulphur dioxide into the stratosphere is one possible idea. However, depending on the latitude, high emission rates can lead to very low transmissions from the perspective of a typical satellite solar occultation instrument, leading to the so-called zero transmission problem. Consequently, it is highly unlikely that a physically meaningful retrieval of the stratospheric aerosol extinction profiles is possible, depending on the latitude and wavelength. The current study analyses, using MAECHAM5-HAM and SCIATRAN, continuous injections of $30\,\text{Tg}\,\text{S}\,\text{yr}^{-1}$ as a hypothetical large-scale stratospheric aerosol injection scenario. For this purpose, sulphur dioxide was continuously injected at an altitude of $60\,\text{hPa}$ ($\approx 19\,\text{km}$) into one grid box ($2.8°\text{x}\,2.8°$) centred on the Equator at $121°\,\text{E}$. Specifically, it is investigated which wavelengths, depending on the latitude, are necessary for plausible aerosol extinction profile retrievals. While a wavelength of $520\,\text{nm}$ is insufficient for the retrieval for $5°\,\text{N}$, the opposite can be concluded for $75°\,\text{N}$ and $75°\,\text{S}$. For the latitudes $45°\,\text{N}$ and $45°\,\text{S}$, a wavelength of at least $1543\,\text{nm}$ is necessary. In contrast, $1900\,\text{nm}$ is sufficient for $15°\,\text{N}$ and $15°\,\text{S}$, as well as $5°\,\text{N}$. Simulation results for an emission rate of $10\,\text{Tg}\,\text{S}\,\text{yr}^{-1}$ show that a minimum wavelength of $1543\,\text{nm}$ is already sufficient for $5°\,\text{N}$. The results emphasize that the zero transmission problem does not mean that solar occultation measurements are entirely useless. Consistent with expectations, a longer wavelength is required for the latitude range of and near the injection. These findings are therefore also relevant for satellite solar occultation measurements after major volcanic eruptions.

## 1 Introduction

Geoengineering, also referred to as climate engineering, encompasses specific techniques and methods that have been proposed to mitigate the consequences of climate change (e.g., Crutzen, 2006). Stratospheric aerosol injections (SAI), i.e the injection of, for example, sulphur dioxide into the stratosphere, are one idea in the field of solar radiation management (SRM) to mitigate the effects of global warming (Budyko, 1977; Crutzen, 2006). Other ideas include the injection of sulphuric acid ($H_2SO_4$) (e.g., Janssens et al., 2020; Weisenstein et al., 2022) or carbonyl sulphide (COS) (e.g., Quaglia et al., 2022) as well as solid materials




such as alumina ($Al_2O_3$) or calcite ($Ca_3CO_3$), as these particles absorb less terrestrial infrared radiation (e.g., Vattioni et al., 2023).

Possible future SAI applications above a certain size can probably be observed with satellite occultation instruments. The
SAGE III/ISS instrument (Stratospheric Aerosol and Gas Experiment III), mounted on the International Space Station (ISS) is a currently active satellite solar occultation instrument, which measures the attenuation of the solar radiation due to scattering and absorption of atmospheric components such as aerosols, ozone, nitrogen dioxide and water vapour. SAGE III/ISS observes around 15 sunrises and 15 sunsets in 24 h and covers a possible latitude range between 70° S and 70° N (NASA, 2022). The instrument has nine spectral channels with aerosols as target species. These specific wavelengths are 384, 448, 520, 601, 676,
755, 869, 1021, and 1543 nm (NASA, 2022).

A previous study showed that it is possible to detect stratospheric aerosols formed from continuous emissions of 1 and 2 Tg S yr$^{-1}$ in the quasi-steady-state phase from the perspective of a typical satellite solar occultation instrument, taking into account the natural variability (Lange et al., 2025). However, 1 and 2 Tg S yr$^{-1}$ are comparatively small emission rates in the context of possible SAI applications. Depending on the progression of climate change and increasing damage and costs, as
well as future goals regarding the reduction of the global mean surface temperature of the Earth, larger to large-scale SAI applications might be considered.

A problem that was already highlighted by the eruption of Mt. Pinatubo in June 1991, which injected about 20 Tg $SO_2$ into the stratosphere, was the occurrence of measurement gaps in areas with high aerosol loadings (e.g., Robock, 2000) due to the resulting low transmission ('zero transmission problem') from the perspective of a satellite solar occultation instrument.
This refers to the transmission of the radiation from the Sun through the atmosphere to the instrument. The SAGE II dataset following the Mt. Pinatubo eruption shows gaps in aerosol measurements for the period June – August 1991 in the region 15° S to 20° N below 22 km, which can be attributed to the dense aerosol plume (e.g., Antuña et al., 2003). The corresponding aerosol plume caused so much extinction of the solar signal that no retrievals were possible (e.g., Stenchikov et al., 1998). During the first year after the eruption of Mt. Pinatubo, the SAGE II instrument only provided measurements above $\approx$ 23 km
altitude at wavelengths of 1024 nm and shorter (e.g., Arfeuille et al., 2013). It should be noted, however, that the eruption of Mt. Pinatubo serves as a natural analogue here, but does not represent continuous emissions. Continuous emissions lead to much lower injection amounts per time compared to a volcanic eruption with the same amount injected.

The aim of this study is to investigate the so-called zero transmission problem using MAECHAM5-HAM simulations and the SCIATRAN radiative transfer model. More specifically, the question is which wavelengths, depending on the latitude, are
necessary for a physically meaningful stratospheric aerosol extinction profile retrieval result, assuming continuous tropical emissions of 30 Tg S yr$^{-1}$ (radiative forcing of about - 4 W/m$^2$ (Niemeier and Timmreck, 2015)), i.e. hypothetical large-scale SAI deployments. In this context, 'physically meaningful' means that not only a retrieval result exists, but that it is also close to the true profile (i.e. based on the MAECHAM5-HAM simulation).

For achieving this, MAECHAM5-HAM simulations for the SRM scenario of 30 Tg S yr$^{-1}$ were used. The simulations
provided vertical profiles of aerosol extinction coefficients at different wavelengths for an altitude range of 10 — 27 km. The aerosol extinction coefficient profiles were used for the transmission calculations with SCIATRAN from the perspective of



a typical solar occultation instrument, which were then used for the aerosol extinction profile retrievals using SCIATRAN. Although the following study analyses the zero transmission problem for a typical satellite solar occultation instrument like SAGE III/ISS, the SAGE retrieval algorithm was not used.

The paper is structured as follows. Section 2 provides an overview of the relevant methods, such as MAECHAM5-HAM and SCIATRAN. Section 3 presents the results, followed by the discussion and conclusions.

## 2   Methodology

### 2.1   MAECHAM5-HAM

Sulphate aerosols were simulated using the ECHAM general circulation model (GCM) in a high-top version (middle atmo-
sphere, MA; (Giorgetta et al., 2006)), extending up to a pressure of 0.01 hPa ($\approx$ 80 km) and comprising 95 vertical levels. The horizontal resolution corresponds to a spectral truncation at wavenumber 63 (T63), equivalent to approximately 1.8° x 1.8°. The prognostic modal aerosol microphysical Hamburg Aerosol Model (HAM, Stier et al. (2005)) was interactively coupled to MAECHAM. HAM calculates the formation of sulphate aerosols, including nucleation, accumulation, condensation, coagulation and removal by sedimentation and deposition. We adapted HAM for stratospheric conditions by implementing a
simple stratospheric sulphur chemistry scheme above the tropopause (Timmreck, 2001; Hommel et al., 2011) and by changing the mode setup, especially the width of the mode bins ($\sigma$) (Kokkola et al., 2009; Niemeier et al., 2009; Niemeier and Timm-reck, 2015). Nucleation includes collision processes for high sulphur loads and adaptations to low stratospheric temperatures (Määtänen et al., 2018).

The simulations were performed with prescribed sea surface temperatures (SSTs) and sea ice, set to to climatological values
(Hurrell et al., 2008), averaged over the AMIP (Atmospheric Model Intercomparison Project) period 1950 to 2000. The model is running freely, no nudging of meteorological values is applied. It calculates the dynamical processes following the equations in the GCM. The quasi-biannual oscillation is also generated in the model and not nudged.

The artificial stratospheric sulphur layer forms from continuous $SO_2$ injections of 30 Tg S yr$^{-1}$. The injection was continuous at an altitude of 60 hPa ($\approx$ 19 km) into one grid box 2.8° x 2.8° centred on the Equator at 121° E.
In the following, the data from the quasi-steady-state phase, an average over 3 years, is analysed. The model output was interpolated to provide aerosol extinction coefficients at different wavelengths for an altitude range of 10 – 27 km, in 1 km steps. The wavelength are as follows: 500, 550, 825, 1050, 1585, 1888, 2250, 2645, 3165 and 3730 nm. In the subsequent analysis, the latitudes -75, -45, -15, 5, 15, 45 and 75° N were examined.

### 2.2   SCIATRAN

For the radiative transfer simulations and aerosol extinction coefficient profile retrievals, the SCIATRAN radiative transfer model version 4.7 was used (Rozanov et al., 2014). SCIATRAN was developed by the Institute of Environmental Physics at the University of Bremen, Germany (Rozanov et al., 2014).





Based on the aerosol extinction coefficients from the MAECHAM5-HAM simulations, aerosol extinction coefficients at the wavelengths relevant for this study were determined using Ångström parameterisation. These aerosol extinction coefficients are used as input for the calculation of the corresponding transmission values from the perspective of a typical solar occultation instrument using SCIATRAN. The calculated transmission values were then used for the retrieval of the corresponding aerosol extinction profiles with SCIATRAN. In the following, a more detailed description of the methods is provided.

The methods described below are similar to the approach used in Lange et al. (2025).

### 2.2.1  Calculation of transmission values

Based on the aerosol extinction coefficients from the MAECHAM5-HAM simulations, the corresponding transmission values from the perspective of a typical satellite occultation instrument were determined using SCIATRAN. Prior to this, the Ångström parameterisation was used to calculate the aerosol extinction coefficients for the wavelengths relevant here from those at the wavelengths used in MAECHAM5-HAM (see Sect. 2.1). Demonstrated in the following for the example of 520 nm (Ångström, 1929):

$$k_{520} = k_{500} \left( \frac{520}{500} \right)^{-\alpha} \tag{1}$$

with:

$$\alpha = - \frac{\ln \left( \frac{k_{550}}{k_{500}} \right)}{\ln \left( \frac{550}{500} \right)} \tag{2}$$

with $k_{500}$ and $k_{550}$ as the given aerosol extinction coefficients at 500 and 550 nm.

In the transmission modelling mode, also called solar occultation mode, the direct solar radiation transmitted through the spherical Earth's atmosphere is simulated. The simulations performed here considered also atmospheric refraction. In addition, vertical profiles of temperature, pressure and trace gases required for the simulations were taken from the implemented climatological database, which is based on a 3-D chemical transport model (Sinnhuber et al., 2003).

Table 1 shows further relevant input parameters for the calculation of transmission values in SCIATRAN. The input parameters are defined to correspond to an imaginary satellite occultation instrument with a satellite altitude of 400 km.

**Table 1.** Input parameters for the calculations of the transmission values with SCIATRAN.

| Parameter | Setting |
|---|---|
| Height grid | 0 - 100 km, 1 km steps |
| Tangent height grid | 10 - 60 km, 2 km steps |
| Vertical field of view | $0.0083°$ |
| Trace gases | $H_2O$, $O_2$, $N_2O$, $NO_3$, $NO_2$, $CO_2$, $O_3$, $SO_2$, $CH_4$, CO |
| Total ozone column (TOC) | 300 DU (Dobson unit) |





The results of the simulations with SCIATRAN are transmission values at 520, 1543 and 1900 nm for tangent heights of 10 to 60 km, in 2 km steps. Although a larger wavelength range was analysed (compare Sect. 2.1), not all wavelengths are shown below in accordance with the objective of this study.

### 2.2.2 Aerosol extinction coefficient profile retrievals

For the retrieval of the aerosol extinction coefficient profiles, the non-freely available retrieval algorithm in SCIATRAN 4.7
was used (Rozanov et al., 2011). The retrieval technique is the regularised inversion with the optimal estimation method. The linearised inverse problem is expressed as follows:

$$y = F(x_a) + K(x - x_a) \tag{3}$$

with $y$ as the measurement vector, containing the logarithms of the transmission values, $F$ as the radiative transfer operator, $x_a$ as the apriori state vector, which remains constant over the iteration steps, $K$ as the weighting function matrix, and $x$ as the
state vector to be retrieved. The apriori state vector $x_a$ contains a background aerosol extinction coefficient profile ($0\,\mathrm{Tg\,S\,yr^{-1}}$) obtained from the MAECHAM5-HAM simulations, which was scaled so that it differs by at least one order of magnitude from the true profile (MAECHAM5-HAM simulations with $30\,\mathrm{Tg\,S\,yr^{-1}}$). Since the apriori is the best available estimate of the true solution and the expected order of magnitude should be reasonable, the scaling of the background profile undertaken here is consistent within the scope of this study in order to ensure a physically plausible initial estimate.

The approximate solution of the inverse problem is determined by minimising the following expression:

$$\|F(x_a) + K(x - x_a) - y\|^2_{S_\epsilon^{-1}} + \|(x - x_a)\|^2_{S_a^{-1}} \tag{4}$$

here with $S_\epsilon$ as the noise covariance matrix and $S_a$ as the apriori covariance matrix. The Gauss–Newton iterative approach is used to formulate the solution for each iteration step:

$$x_{i+1} = x_a + \left(K_i^T S_\epsilon^{-1} K_i + S_a^{-1}\right)^{-1} K_i^T S_\epsilon^{-1} \left(y - F(x_i) + K_i(x_i - x_a)\right) \tag{5}$$

More information on the SCIATRAN retrieval algorithm can be found in Rozanov et al. (2011), Sect. 3.4.2.

  The following Tab. 2 shows the relevant input parameters for the aerosol extinction coefficient profile retrievals with SCIA-TRAN. The settings regarding the height grid, tangent height grid, vertical field of view and total ozone column are the same as for the forward simulations (compare Tab. 1). The retrievals were restricted to the altitude range of the provided data (compare Sect. 2.1), in this case from 10 to 27 km. The defined signal-to-noise-ratio (SNR) varies depending on the tangent height (TH),
assuming constant noise:

$$\mathrm{SNR(TH)} = \mathrm{SNR_{max}} \cdot \frac{T(\mathrm{TH})}{T_{max}} \tag{6}$$

where $T_{max}$ is the maximum transmission value ($\approx 1$ at TH = 60 km), $\mathrm{SNR_{max}}$ the corresponding maximum SNR (1000 (e.g. Meyer et al. (2005)) at TH = 60 km) and $T_{TH}$ the transmission value at the tangent height TH.





**Table 2.** Relevant input parameters for the aerosol extinction coefficient profile retrievals with SCIATRAN.

| Parameter | Setting |
|---|---|
| A priori variance | 30 % |
| Convergence criterion | 2 % |
| Signal-to-noise-ratio (SNR) | Tangent height dependent (see Eq. 6) |

The results of the retrievals with SCIATRAN are aerosol extinction coefficient profiles at 520, 1543 and 1900 nm for the relevant latitudes.

## 2.3 Error analysis

Assuming random, statistically independent error sources and a linear dependence of the derived aerosol extinction coefficients on the parameters, the following approach was used for the error estimation:

$$\sigma^2_{\text{total}} = \sigma^2_{\text{Total ozone column}} + \sigma^2_{\text{Temperature and pressure}} + \sigma^2_{\text{Temperature}} + \sigma^2_{\text{Pressure}} + \sigma^2_{\text{Pointing error}} + \sigma^2_{\text{Noise}} \tag{7}$$

Each term in Eq. 7 represents individual errors in aerosol extinction caused by incorrect knowledge or uncertainties of relevant input parameters, e.g. pointing, pressure, temperature and total ozone (compare Tab. 3). These individual errors represent relative differences $r$ (Eq. 8) between the retrieved aerosol extinction profiles based on the reference setting and the modified setting (compare Tab. 3). Here, ref is the retrieved aerosol extinction profile based on the reference settings and $x$ the retrieved aerosol extinction profile based on the modified settings (Eq. 8). The noise error was obtained from the noise covariance matrix.

**Table 3.** Reference and modified settings for the error analysis.

| Parameter | Reference setting | Modified setting |
|---|---|---|
| Total ozone column (TOC) | 300 DU | + 2 % (e.g., Garane et al., 2019) |
| Temperature and pressure (constant air density) | - | + 2 K (e.g., Nowlan et al., 2007; Langland et al., 2008) |
| Pressure | - | + 2 % (e.g., Nowlan et al., 2007; Langland et al., 2008) |
| Temperature | - | + 2 K (e.g., Nowlan et al., 2007; Langland et al., 2008) |
| Pointing error | - | + 100 m tangent height grid (e.g., Bramstedt et al., 2012) |

$$r = \frac{x - \text{ref}}{\text{ref}} \cdot 100\% \tag{8}$$

Figure A1 in the appendix illustrates an excerpt of the relative differences $r$ for the retrieved aerosol extinction profiles at 520 nm and 75° S.



# 3 Results and discussion

Figure 1 shows aerosol extinction coefficients at 550 nm (1/km) for January (Jan) based on the MAECHAM5-HAM simulations for the continuous injection of $30\,\mathrm{Tg\,S\,yr^{-1}}$ (upper panel) and the corresponding aerosol optical depth (AOD) at 550 nm (lower panel).

**Figure 1.** Upper panel: Aerosol extinction coefficients at 550 nm (1/km) for January (Jan) based on the MAECHAM5-HAM simulations for the continuous injection of $30\,\mathrm{Tg\,S\,yr^{-1}}$. Lower panel: Corresponding aerosol optical depth (AOD) at 550 nm.



The maximum in AOD near the Equator (lower panel of Fig. 1) is due to the continuous injection of $30\,\mathrm{Tg\,S\,yr^{-1}}$ in this region (compare Sect. 2.1). Figure 1, i.e. the latitude dependence of the aerosol extinction coefficients and the corresponding AOD, highlights the fact that the minimum wavelength required to obtain a physically plausible retrieval result depends on the specific latitude. Note that this depends not only on the AOD but also on the vertical profile of the aerosol extinction coefficients. Nevertheless, the AOD (550 nm) for the corresponding latitudes is given below for better contextualisation.

Panel (a) of Fig. 2 shows as an example the aerosol extinction profile retrieval result for $5°\,\mathrm{N}$ if performed for a wavelength of 520 nm. With the retrieved aerosol extinction profile in blue, the apriori profile in black and the true profile (MAECHAM5-HAM simulation result) in orange. Panel (b) of Fig. 2 shows the averaging kernels resulting from the retrieval of the aerosol extinction profile at 520 nm and $5°\,\mathrm{N}$ (blue line in panel (a)).

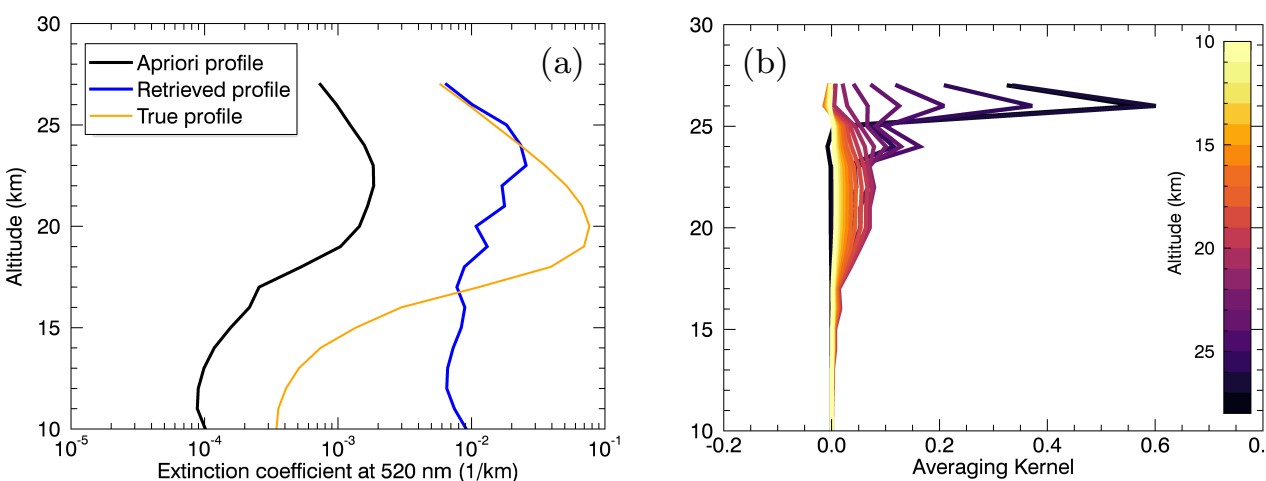

**Figure 2.** (a) Retrieved aerosol extinction profile (blue line), apriori profile (black line) and true profile (MAECHAM5-HAM simulation result) (orange line) at 520 nm and $5°\,\mathrm{N}$. (b) Averaging kernels resulting from the retrieval of the aerosol extinction profile at 520 nm and $5°\,\mathrm{N}$ (blue line in panel (a)).

As expected, the retrieved aerosol extinction profile (blue line in panel (a) of Fig. 2) shows below $\approx 17\,\mathrm{km}$ a behaviour similar to the apriori profile (black line in panel (a) of Fig. 2), above $\approx 25\,\mathrm{km}$ a good agreement with the true profile (orange line in panel (a) of Fig. 2), and in between oscillations of the profile. Overall, the retrieved aerosol extinction profile is not in good agreement with the true profile. Consistent with these findings, the averaging kernels (panel (b) of Fig. 2) show very small values close to zero below about 24 km. At these altitudes, the retrieval exhibits little to no sensitivity to the measurement.

Furthermore, Fig. 3 illustrates, on the one hand, the correspondingly low transmission values from the perspective of a typical solar occultation instrument, which are $\approx 10^{-14}$ at a minimum (TH = 18 km), based on the true vertical profile of the aerosol extinction coefficients at 520 nm and $5°\,\mathrm{N}$ (orange line in panel (a) of Fig. 2). On the other hand, it highlights the low agreement between the retrieved and true profile. The black line in panel (a) of Fig. 3 represents the true transmission values



based on the true aerosol extinction profile (MAECHAM5-HAM simulation), and the blue line the transmission values based on the retrieved aerosol extinction profile. Panel (b) of Fig. 3 illustrates the corresponding relative residuals of the transmission values in %.

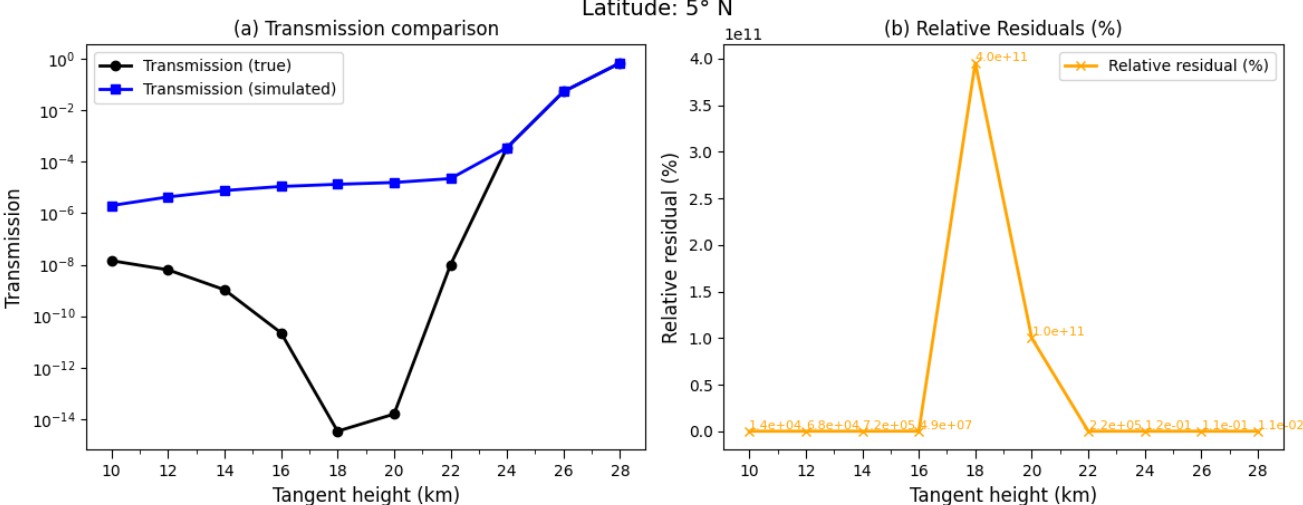

**Figure 3.** (a) Comparison of true (black line; based on true aerosol extinction profile (MAECHAM5-HAM simulation)) and simulated (blue line; based on retrieved aerosol extinction profile) transmission values at 520 nm. (b) Corresponding relative residuals in %. Latitude: 5° N.

In summary, a wavelength of 520 nm is not sufficient for the aerosol extinction coefficient retrieval at 5° N. Figure 4 provides an overview of the retrieval results, i.e. the retrieved aerosol extinction profiles, when the corresponding retrieval for 5° N is

performed at 520 nm (same as panel (a) of Fig. 2) (a) and at additional larger wavelengths of 1543 nm (b), 1800 nm (c) and 1900 nm (d). Note that the wavelengths of 520 and 1543 nm were chosen as these are two of the wavelengths at which the currently active satellite solar occultation instrument SAGE III/ISS, targets aerosols. 1543 nm is the largest of these wavelengths targeting aerosols (NASA, 2022).





**Figure 4.** Retrieved aerosol extinction profile (blue line), apriori profile (black line) and true profile (MAECHAM5-HAM simulation result) (orange line) for 5° N and 520 nm (a), 1543 nm (b), 1800 nm (c) and 1900 nm (d).

The panels (a) – (d) of Fig. 4 clearly show that with increasing wavelength, the retrieved aerosol extinction profile gets closer to the true profile with the best agreement at 1900 nm (especially at the maximum, compared to 1800 nm). This means that, under the assumptions made here, a wavelength of at least 1900 nm is required for the aerosol extinction profile retrieval at 5° N (AOD ≈ 0.45 at 550 nm). Therefore, panel (d) also shows the retrieved profile including the total errors (blue dashed lines), as described in Sect. 2.3.





Corroborating this finding, panel (b) of Fig. 5 presents a maximum value of the relative residuals between the true (based on

the true aerosol extinction profile (MAECHAM5-HAM simulation)) and simulated (based on the retrieved aerosol extinction

profile) transmission values of 1.4 % (TH = 22 km).

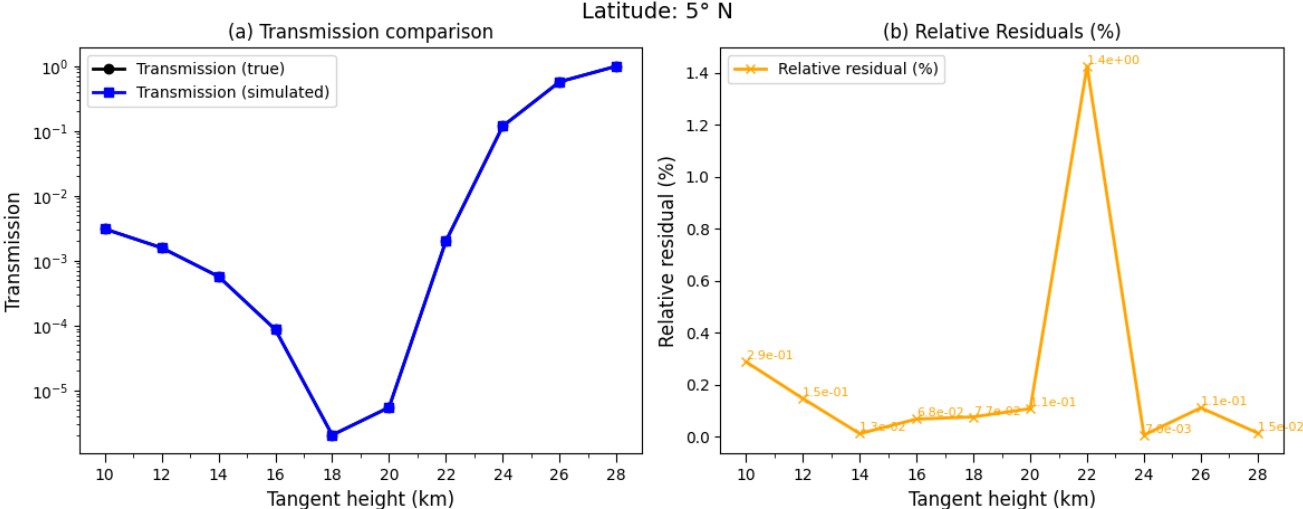

**Figure 5.** Same as Fig. 3 but for 1900 nm. Note that the graphs in panel (a) are nearly identical and therefore only one is visible.

Although 520 nm is not sufficient for the aerosol extinction profile retrieval at 5° N, a different conclusion can be drawn for

75° S (AOD ≈ 0.35 at 550 nm) (panel (a) of Fig. 6) and 75° N (AOD ≈ 0.24 at 550 nm) (panel (b) of Fig. 6). The blue dashed

lines show the corresponding total errors (compare Sect. 2.3).





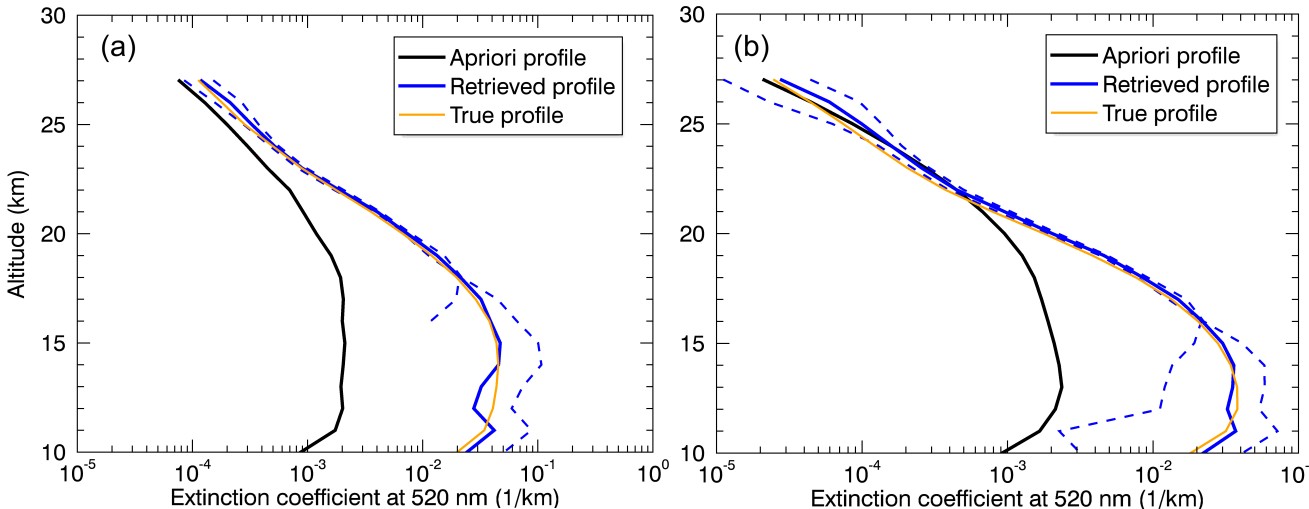

**Figure 6.** Retrieved aerosol extinction profile (blue line) including total error (blue dashed lines), apriori profile (black line) and true profile (MAECHAM5-HAM simulation result) (orange line) for 75° S (a) and 75° N (b) both for 520 nm.

In addition, Fig. 7 shows a comparison of the transmission values at 520 nm (a) and the corresponding relative residuals (b) for 75° N. The comparison for 75° S can be found in the appendix (Fig. A2). The relative residuals show the highest value of more than 100 % at TH = 12 km, which then decrease to less than 1 % (compare panel (b) of Fig. 7).

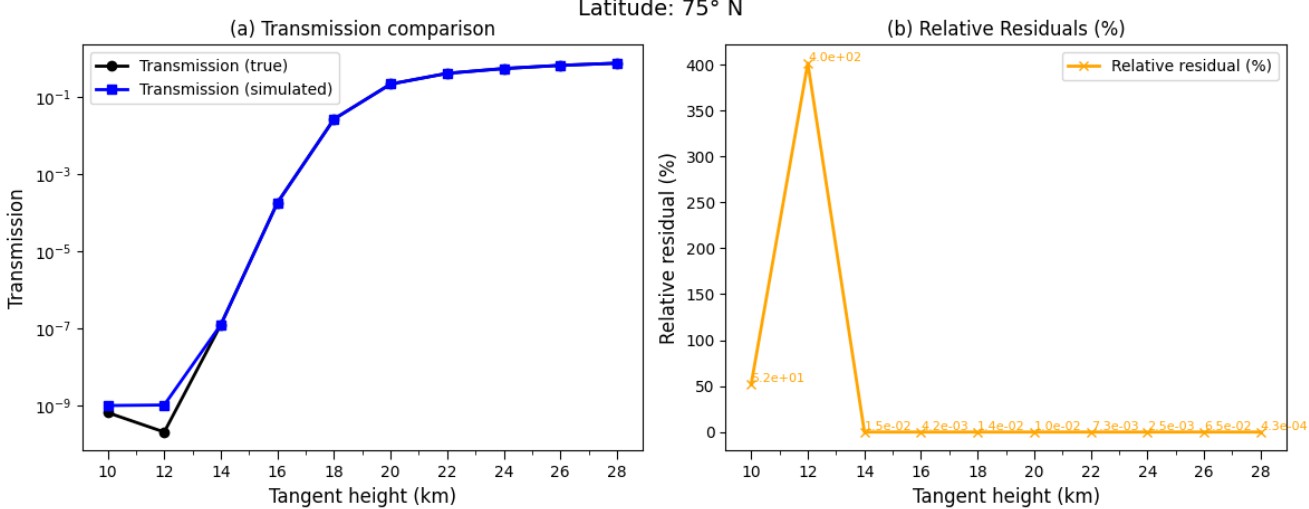

**Figure 7.** Same as Fig. 3 but for 75° N.





This means that, under the assumptions made here, a wavelength of at least 520 nm is required for the aerosol extinction profile retrieval for 75° S (AOD ≈ 0.35 at 550 nm) and 75° N (AOD ≈ 0.24 at 550 nm). In contrast, Fig. 8 illustrates the
retrieved vertical profiles of the aerosol extinction coefficients at 45° S (a) and 45° N (b) for 1543 nm, which are in good agreement with the true profiles.

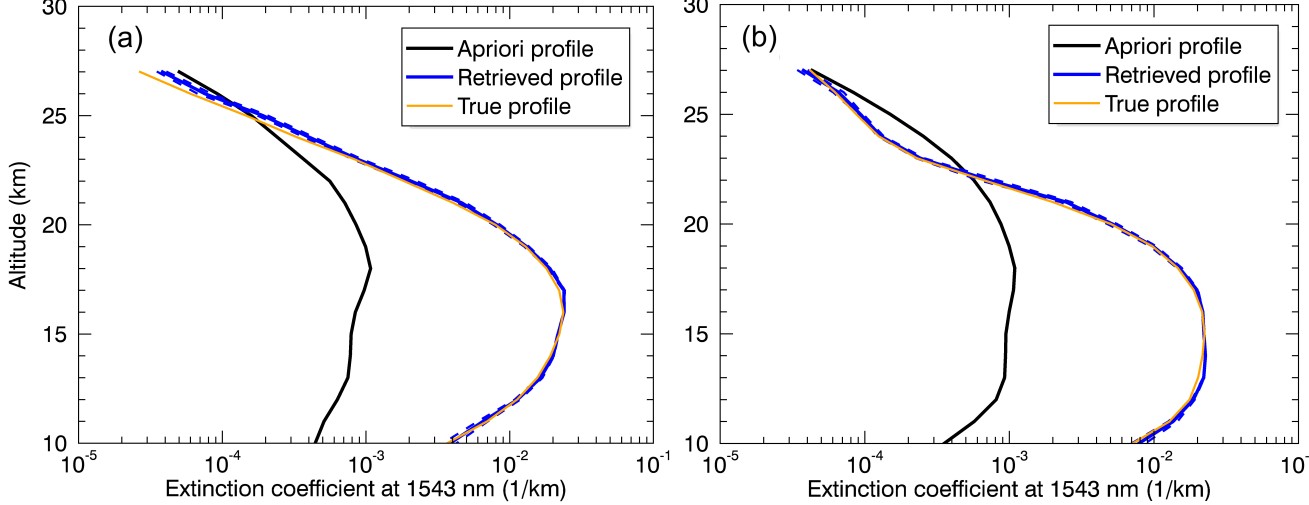

**Figure 8.** Retrieved aerosol extinction profile (blue line) including total error (blue dashed lines), apriori profile (black line) and true profile (MAECHAM5-HAM simulation result) (orange line) for 45° S (a) and 45° N (b) both for 1543 nm.

In line with these results, the corresponding transmission values (panel (a) of Fig. 9) are in very good agreement, with relative residuals of less than 1 % (panel (b) of Fig. 9). The illustration for 45° S can be found in the appendix (Fig. A3).



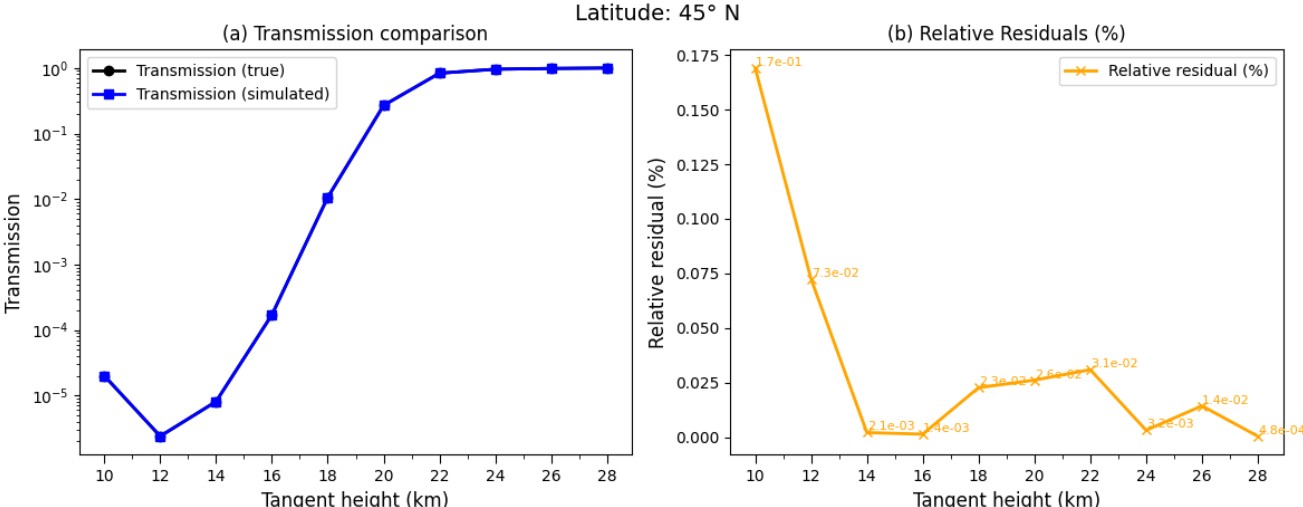

**Figure 9.** Same as Fig. 3 but for 45° N and 1543 nm.

It can therefore be concluded, taking into account the assumptions made here, that with continuous injections of 30 Tg S yr$^{-1}$,
a wavelength of at least 1543 nm is required for the aerosol extinction profile retrieval for latitudes of 45° N (AOD ≈ 0.3 at
550 nm) and 45° S (AOD ≈ 0.3 at 550 nm). The following Fig. 10 shows the aerosol extinction profile retrieval results for
15° S (panel (a)) and 15° N (panel (b)) for a wavelength of 1900 nm. As is the case for 5° N, a wavelength of at least 1900 nm
is required for the retrieval of the vertical profiles of the aerosol extinction coefficients at 15° N (AOD ≈ 0.25 at 550 nm) and
15° S (AOD ≈ 0.35 at 550 nm).





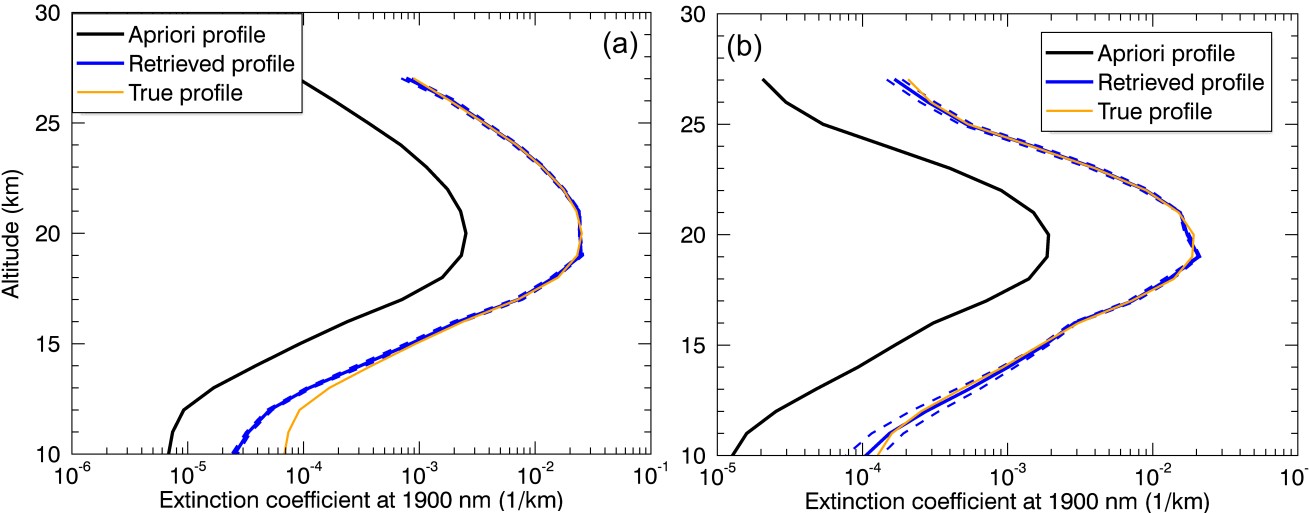

**Figure 10.** Retrieved aerosol extinction profile (blue line) including total error (blue dashed lines), apriori profile (black line) and true profile (MAECHAM5-HAM simulation result) (orange line) for 15° S (a) and 15° N (b) both for 1900 nm.

The corresponding transmission values from the perspective of a typical satellite solar occultation instrument show the greatest difference at TH = 20 km, with 29 % for 15° N (Fig. 11). The relative residuals for 15° S are less than 1 % ((Fig. A4)).

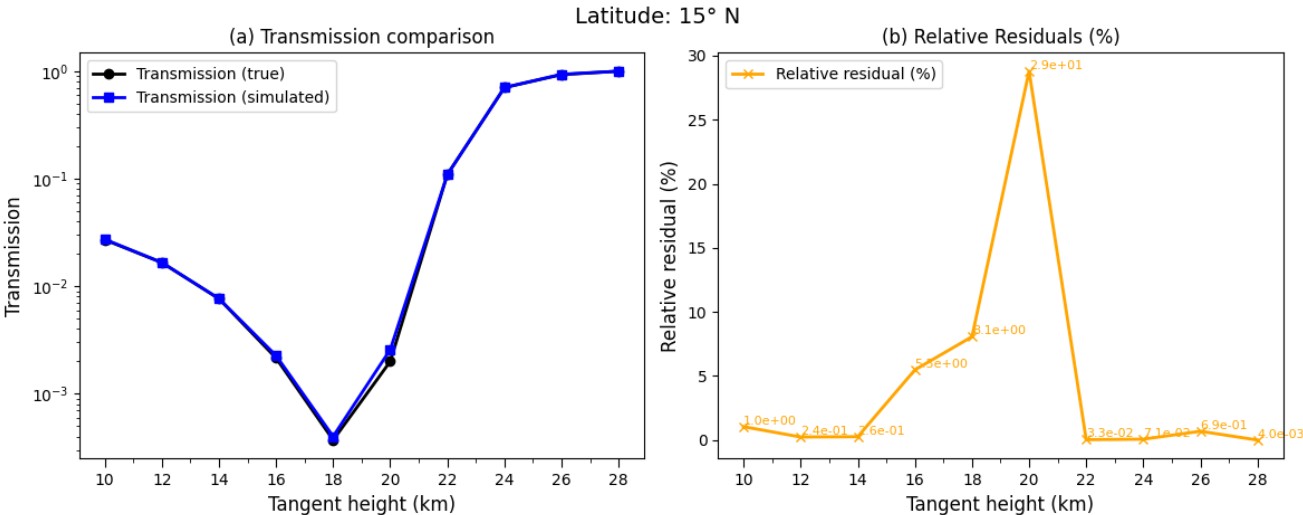

**Figure 11.** Same as Fig. 3 but for 15° N and 1900 nm.

The results shown above demonstrate that with continuous injections of $30\,\mathrm{Tg\,S\,yr}^{-1}$, different minimum wavelengths are required for a physically meaningful retrieval result, depending on the latitude. The wavelength-latitude combinations are as





follows: 520 nm for 75° N and S, 1543 nm for 45° N and S as well as 1900 nm for 15° N and S and 5° N. These findings also
215   indicate that the longest wavelength at which SAGE III/ISS targets aerosols, i.e. 1543 nm, is not sufficient for the aerosol
extinction profile retrievals for 5° N, 15° N and S, i.e. latitudes near the injection, under the assumptions made here. However,
the results show that the zero transmission problem does not mean that solar occultation measurements are entirely useless.

Figure 12 shows the total errors (in %) for the aerosol extinction profile retrievals for these wavelength-latitude combinations
based on the error analysis described above (Sect. 2.3).

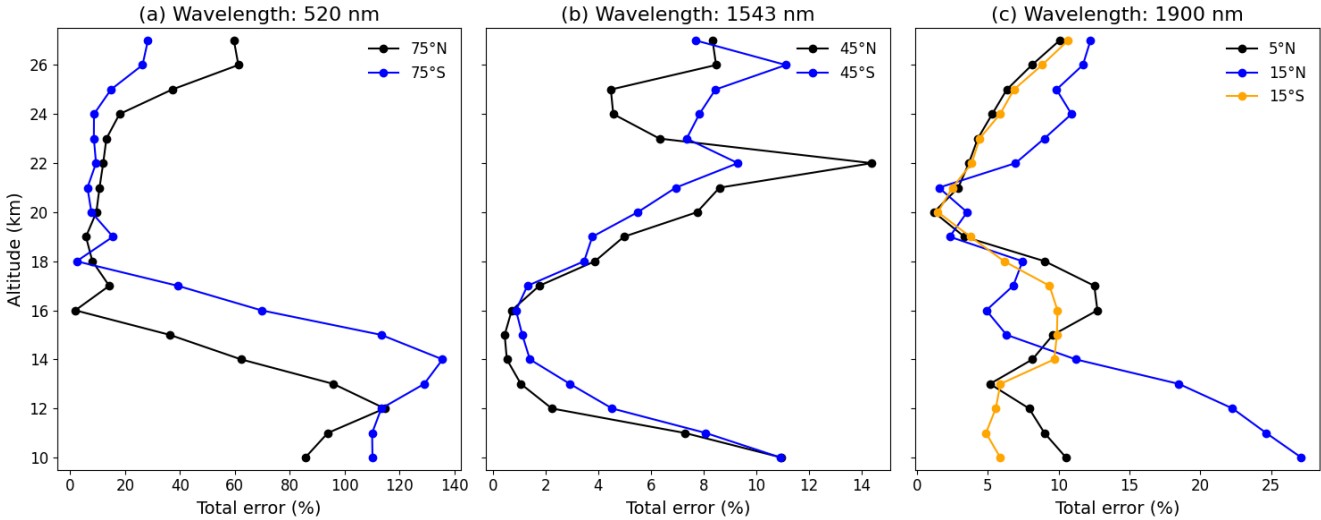

**Figure 12.** Total error in %. (a) 520 nm for 75° N, 75° S, (b) 1543 nm for 45° N, 45° S and (c) 1900 nm for 5° N, 15° N, 15° S.

220   The total errors vary depending on wavelength, latitude and altitude. The total errors at the altitude of the injection, here
60 hPa ($\approx$ 19 km), are the following: 6 % (75° N), 16 % (75° S) (panel (a) of Fig. 12), as well as $\approx$ 4 % (45° N and S) (panel
(b)) and $\approx$ 3 % (15° N and S, 5° N) (panel (c)). Note that the large errors at low altitudes are due to the low aerosol extinction
coefficients at these altitudes (compare, e.g., Fig. 10).

Furthermore, MAECHAM5-HAM simulation results for an emission rate of 10 Tg S yr$^{-1}$ were examined for the latitude
range near the injection. The results show that a minimum wavelength of 1543 nm is already sufficient for 5° N (not shown).

Although 30 Tg S yr$^{-1}$ appears to be a comparatively high emission rate, the upper limit in the context of possible SAI
applications depends, for instance, on the specific goal, such as the aspired temperature reduction and radiative forcing.

## 4   Conclusions

The present study examined which wavelengths, depending on the latitude, are necessary for the aerosol extinction profile
retrievals in the case of continuous tropical injections of 30 Tg S yr$^{-1}$.





While a wavelength of 520 nm is insufficient for the retrieval for 5° N, the opposite can be concluded for 75° N and 75° S. For the latitudes 45° N and 45° S, a wavelength of at least 1543 nm is necessary. In contrast, 1900 nm is sufficient for 15° N and 15° S, as well as 5° N. Consistent with expectations, a longer wavelength is required for the latitude range of and near the injection, in this case at least 1900 nm.

The results also emphasise that the zero transmission problem does not mean that solar occultation measurements are entirely worthless. Depending on the latitude (see above), the already operating satellite solar occultation instrument SAGE III/ISS could probably also provide aerosol measurements in the relevant altitude range, assuming continuous emissions of $30\,\text{Tg S yr}^{-1}$.

    The results of this study also confirm the generally accepted idea that, in the case of very high emissions, such as $30\,\text{Tg S yr}^{-1}$,

which lead to extremely low transmission values (here about $10^{-14}$ at 520 nm at a minimum – from the perspective of a typical solar occultation instrument), the appropriate approach is to use longer wavelengths for aerosol measurements. Therefore, the results are also relevant for measurements following major volcanic eruptions.

*Code and data availability.* SCIATRAN can be downloaded via the following link: https://www.iup.uni-bremen.de/sciatran/.

**Appendix A**

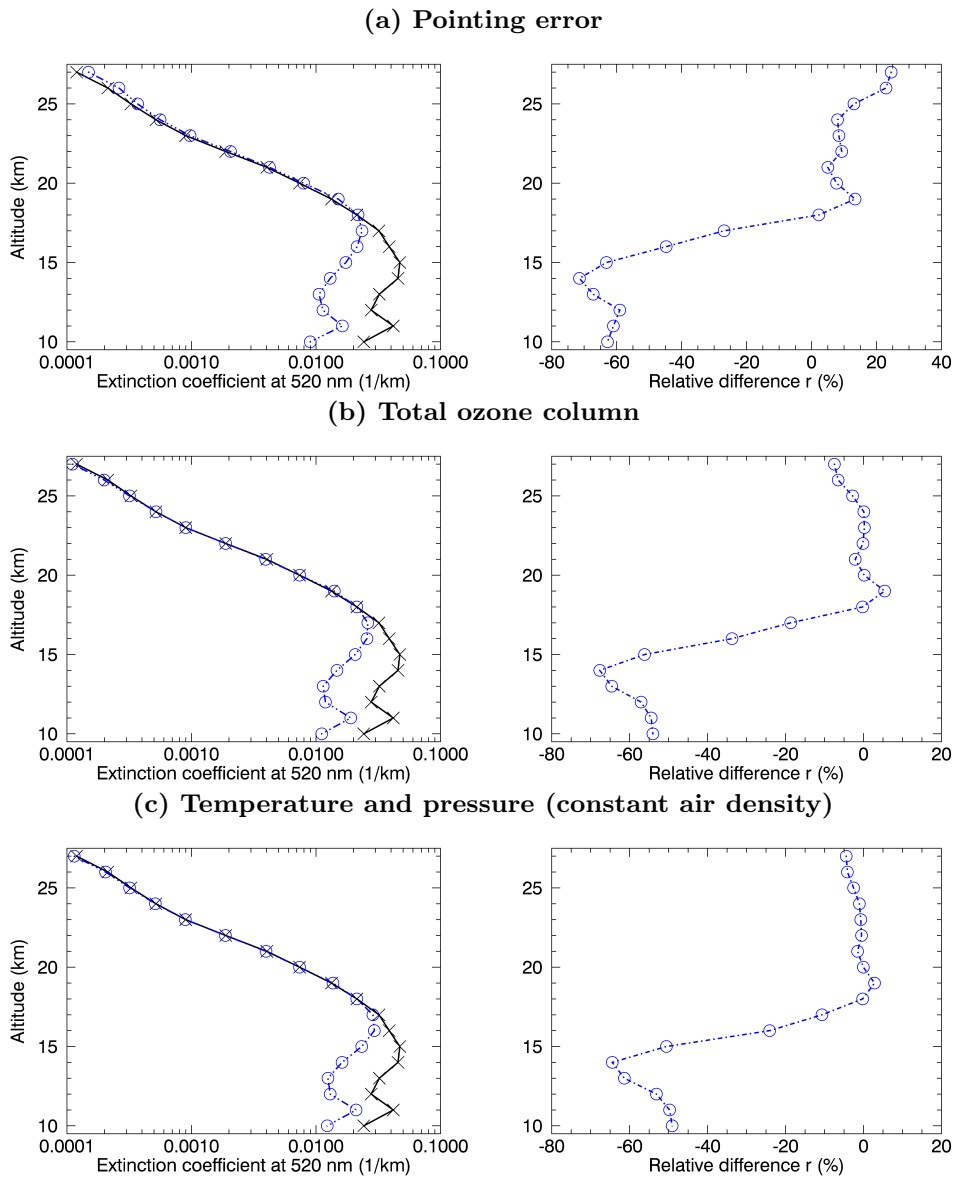

**Figure A1.** Left column: Retrieved aerosol extinction profiles at $520\,\text{nm}$ (1/km) with reference settings (black line) and modified settings (blue line) for $30\,\text{Tg}\,\text{S}\,\text{yr}^{-1}$, $75°\,\text{S}$. Right column: Corresponding relative difference $r$. Both for (a) Pointing error, (b) Total ozone column and (c) Temperature and pressure (constant air density).




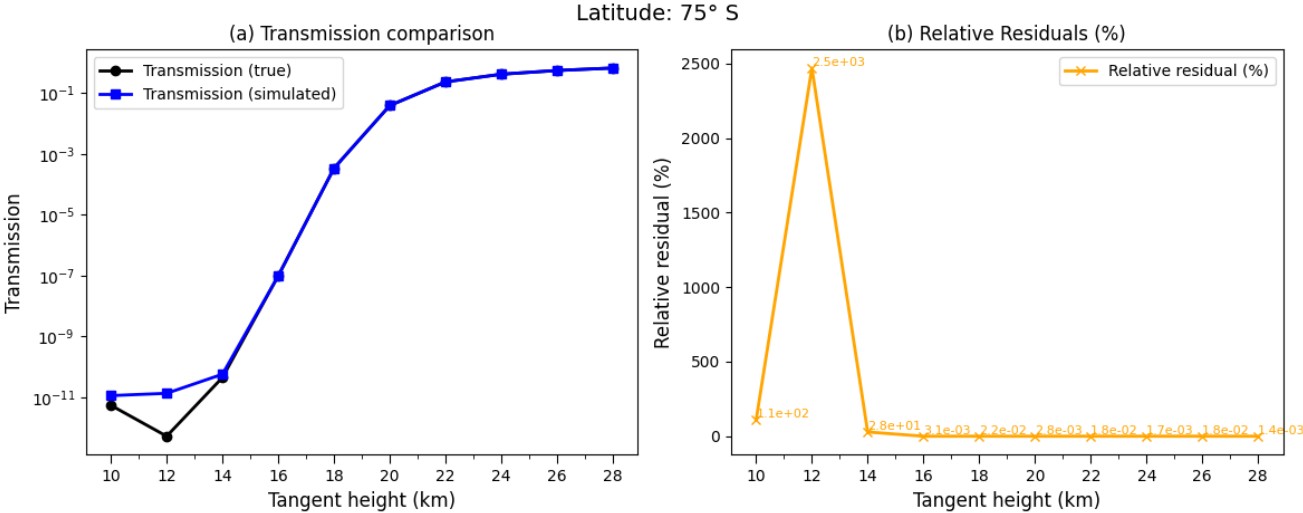

**Figure A2.** (a) Comparison of true (black line; based on true aerosol extinction profile (MAECHAM5-HAM simulation)) and simulated (blue line; based on retrieved aerosol extinction profile) transmission values at 520 nm. (b) Corresponding relative residuals in %. Latitude: 75° S.





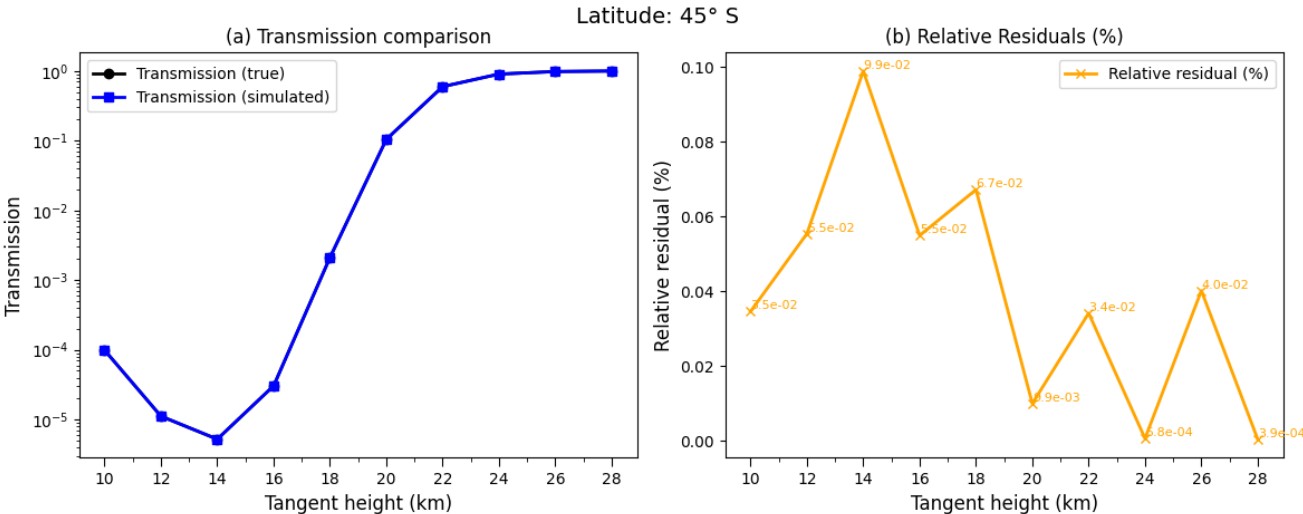

**Figure A3.** Same as Fig. A2 but for 45° S and 1543 nm.

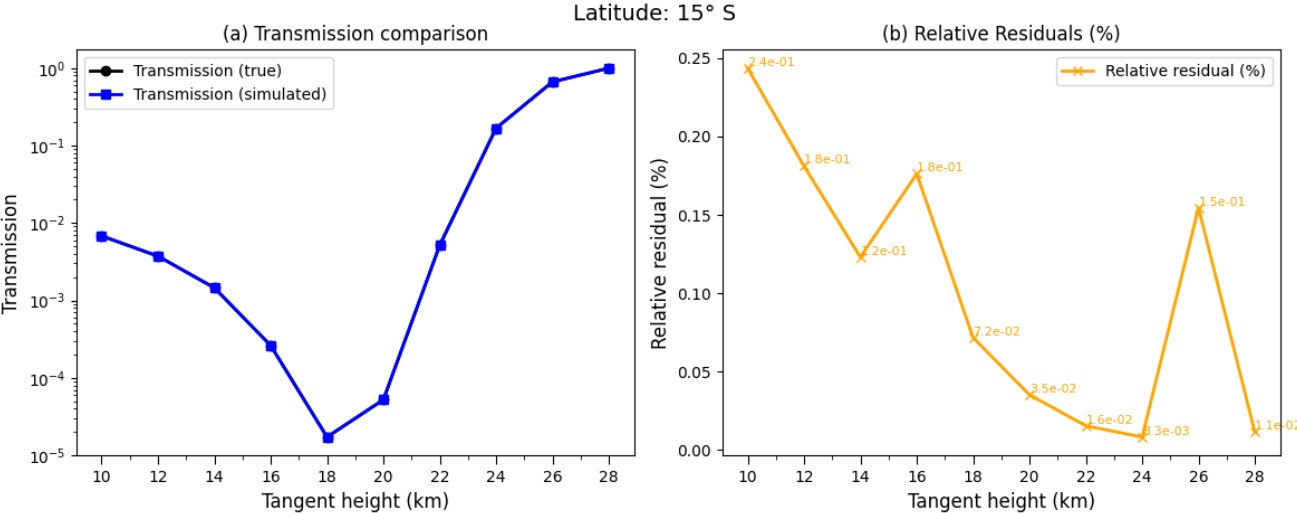

**Figure A4.** Same as Fig. A2 but for 15° S and 1900 nm.



*Author contributions.* AL carried out the transmission calculations and retrievals using SCIATRAN with guidance by CvS and AR. UN performed the MAECHAM5-HAM model simulations and wrote the MAECHAM5-HAM methodology subsection. AL wrote an initial version of the paper. All authors discussed, edited and proofread the paper.

*Competing interests.* Christian von Savigny is an editor of Atmospheric Measurement Techniques.

*Acknowledgements.* We are indebted to the Institute of Environmental Physics at the University of Bremen, in particular to Alexei Rozanov,
for the access to the SCIATRAN retrieval algorithm. This study was enabled by the collaborations within the DFG research unit Volimpact (FOR 2820, grant no. 398006378).



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
