# Peer review of "Investigating the zero transmission problem in satellite solar occultation measurements in the context of possible stratospheric aerosol injections"

_EGUsphere, 2025_

## Referee Comment (RC1)

**Overview**

The authors present a brief study on the utility of solar occultation measurements under heavy stratospheric aerosol loading caused by continuous injection of 30 Tg/year sulfur (i.e., as part of geoengineering efforts). The authors employed a model to construct the theoretical atmosphere, from which they extracted aerosol loadings, which were then used to estimate the solar transmission through the atmosphere (i.e., simulated a solar occultation observation), followed by a retrieval of extinction profiles. The authors then evaluated the accuracy of these extinction profiles at select wavelengths and select latitudes.

The concept of this study is very interesting, and I am excited to see the paper in its final form. This paper, when complete, will make a substantive contribution to AMT, and fits well within the scope of the journal. However, the paper is unfortunately incomplete in its current form. The evaluation is presented very quickly, often with only brief statements to figures, and lacks the presentation it deserves. Ultimately, this paper leaves the reader pondering many questions; questions that should have been answered in the paper. After reading the paper, I am left with the desire that the authors take more time to better present their work so I can better understand and appreciate it. Such changes should be relatively simple additions, so my recommendation is for publication after major revisions and another round of reviews.

**Major Remarks**

I have 2 major criticisms of the current version of this paper.

First, the authors must improve the clarity of their methodology. As written, I could not reproduce this work without making multiple key assumptions. Some of these will be highlighted in the next section.

Second, the paper lacks a deeper discussion of what their results mean. In Results and Discussion, there are a lot of results and little discussion. Most of what is given to us is along the lines of "we tested this wavelength at this latitude and got Fig. X"; it boils down to a presentation of a series of figures without meaningful discussion.

I raise these concerns for 2 reasons. First, I see great value in this fundamental study. If the authors enhance their discussion then I believe this can be a *great* paper. Second, the authors undoubtedly put a lot of effort into this study as well as a great deal of thought. Don't deprive the reader of everything you learned through this exercise.

**Specific Remarks**

– **page 2, line 35:** What is meant by "larger to large-scale"? Can this be quantified?

– **page 2, lines 40-45:** The discussion of SAGE II observations says the same thing 3 times. Please condense and reword for clarity.

– **page 3, line67:** "The prognostic modal aerosol microphysical Hamburg Aerosol Model..." This is hard to mentally digest. Can this be reworded for clarity.

– **page 3, line 74:** "...set to to climatological..." to "...set to climatological..."

– **page 3, line 82:** "wavelength" to "wavelengths"

– **page 5, line 112:** It is unclear what the authors mean by "in accordance with the objective of this study." As written, it sounds like the objective of the study was to *not* look at all wavelengths. If that is the case, then how can the authors make wavelength recommendations? Please clarify.

– **Section 3:** I assume the authors allowed the model to achieve steady-state conditions. Is this correct? How many years did the model have to run before they started their retrievals?

– **Figure 1, panel B:** While interesting, a line-of-sight (LOS) optical depth would be more informative to occultation observations. This would also show the reader why retrieval below the aerosol peak is problematic for certain wavelengths. Further, breaking this into a multi-panel figure that has LOS OD for all wavelengths used in the current study (LOS OD as a function of altitude and latitude) would communicate how and why different wavelengths perform better at different altitudes and latitudes. This figure will strengthen the authors' arguments later in the paper so I ask the authors to please consider making this addition.

– **page 8, line 161:** "...is given below..." Figure positioning is relative, please provide a definite reference to the figure you want.

– **Figure 3:** What does this figure add? Why would you calculate transmission from your extinction coefficients? This seems unnecessary when the authors can use the extinction coefficients directly to calculate percent difference, which is more meaningful (if transmission is more meaningful, please explain how).

– **Figure 4:** Please correct the x-tick labels in panel (b) to match the other panels.

– **page 12, line 197:** Why do the residuals blow up at 12 km and nowhere else? This is an example of the authors basically reading the figures to the reader without bringing additional insight to the discussion. Please convey to the reader why this happens and why it is significant (or not significant).

– **Figure 8:** Did you try this with shorter wavelengths? What was the result?

– **Figure 11:** Why is the residual so high? From Fig. 1 I expected this to be better than the southern hemisphere, but it is worse; why? Also, why the massive difference between Fig. 11 and Fig. A4 (roughly 2 orders of magnitude)? Please explain.

– **page 16, line 217:** "...does not mean that solar occultation measurements are entirely useless." Please reword for clarity. As written, this is not correct. If there is zero transmission then there is zero signal, which makes the measurements useless.

– **page 16: line 222:** "...due to the low extinction coefficients at these altitudes..." Please reword for clarity. I doubt low extinction is the dominant problem. It is more likely the massive line-of-sight optical depth caused by the thick aerosol layer higher in the atmosphere, which obscures the signal at lower altitudes. I note that a line-of-sight optical depth plots (as suggested above) would easily communicate this.

– **page 16, lines 226-227:** It is unclear what this is intended to communicate or why it is relevant to this study. Please clarify or remove.

– **page 16, lines 229-230:** "The present study examined which wavelengths, depending on the latitude, are necessary for the aerosol extinction profile retrievals in the case of continuous tropical injections of 30 Tg S yr$^{-1}$." This is not correct. Most figures showed data from only 1 wavelength at 1 latitude without evaluating performance at other wavelengths/latitudes. What can we learn about other latitudes outside the 4 you present in the paper? What wavelengths can be used at, say, 30 N? Are there hard cutoffs, in latitude, where 520 nm suddenly fails to yield a viable product and we must then move to 1543 nm? What about 1020 nm, or 755 (the authors were focused on SAGE wavelengths)? Ultimately, I am left knowing very little more than I knew before reading the paper (i.e., longer wavelengths are required to see through denser aerosol layers). In short, we are forced to accept the authors' conclusions without seeing the evidence. Please reinforce your conclusions by "fleshing out" the analysis and giving us evidence for your conclusions.

– **page 17, lines 233-234:** This is more of a side comment, but what the authors say here presents the opportunity for my comment. Yes, but it is also important to know if the atmosphere is in steady state (the authors deny us the benefit of these details!). If not, then the optical depth will continue to change, which may make the 1543 and 1900 nm channels insufficient for the task. Please provide better simulation details.

– **page 17, line 235:** "...does not mean that solar occultation measurements are entirely worthless." I doubt anyone would conclude that, a priori. However, as stated above, if we are in a true zero transmission situation then there is not signal. Please clarify your intent.

– **page 17, lines 236-237:** Please be more precise in your language. You can state specifically what wavelengths are useful at each latitude and altitude range. Please provide this information.

– **General statement:** The authors conclude that a 1900 nm channel is required to get a viable retrieval through the most dense aerosol layers, but they fail to discuss why we wouldn't use solely longer wavelengths (i.e., ≥1900 nm) everywhere/all the time.

– **General statement:** The authors refer to the 1991 Pinatubo eruption (20 Tg S) as a reference point for their assumed 30 Tg/year injection rate. A continuous injection is far different from an acute injection. I would assume that the continual injection would continue to build a reservoir of stratospheric sulfur that exceeds what we would get from a single, rapid, injection of 30 Tg. Can the authors comment on how a continual injection of 30 Tg corresponds to a single eruption (e.g., once steady state is achieved, what size of a volcanic eruption would be required to achieve a comparable stratospheric aerosol load)? This may be more than the authors can offer, so if they cannot provide this information that is fine.

– **General statement (last one!):** One aspect the authors neglected is how coincident volcanic eruptions may amplify the zero transmission issue. For example, how would a relatively modest (VEI 2/3) eruption impact wavelength selection as a function of altitude? If Raikoke, Calbuco, or Puyehue-Cordon Caulle erupted again, would the 520 nm channel still remain robust at higher latitudes? Would a larger tropical eruption provide too much load for 1900 nm to remain useful in the tropics? I see this as exciting future work and look forward to the authors' future publications.